# Prospective Measures of Adherence by Questionnaire, Low Immunosuppression and Graft Outcome in Kidney Transplantation

**DOI:** 10.3390/jcm10092032

**Published:** 2021-05-10

**Authors:** Mathilde Prezelin-Reydit, Valérie Dubois, Sophie Caillard, Anne Parissiadis, Isabelle Etienne, Françoise Hau, Laetitia Albano, Monique Pourtein, Benoît Barrou, Jean-Luc Taupin, Christophe Mariat, Léna Absi, Cécile Vigneau, Virginie Renac, Gwendaline Guidicelli, Jonathan Visentin, Pierre Merville, Olivier Thaunat, Lionel Couzi

**Affiliations:** 1Service de Néphrologie-Transplantation-Dialyse-Aphérèse, Hôpital Pellegrin, CHU de Bordeaux Pellegrin, 33076 Bordeaux, France; pierre.merville@chu-bordeaux.fr (P.M.); lionel.couzi@chu-bordeaux.fr (L.C.); 2AURAD Aquitaine, 2 allée des demoiselles, 33170 Gradignan, France; 3Equipe Biostatistique, INSERM U1219 BPH, 14 rue Léo Saignat, 33000 Bordeaux, France; 4CIRI, INSERM U1111, CNRS UMR5308, Ecole Normale Supérieure de Lyon, Université Claude Bernard Lyon I, 21, avenue Tony Garnier, 69007 Lyon, France; Valerie.Dubois@efs.sante.fr (V.D.); olivier.thaunat@chu-lyon.fr (O.T.); 5HLA Laboratory, French National Blood Service (EFS), 111, rue Elisée-Reclus, 69153 Décines-Charpieu, France; 6Nephrology-Transplantation Department, University Hospital, 1 Place de l’hôpital BP426, CEDEX, 67091 Strasbourg, France; sophie.caillard@chru-strasbourg.fr; 7INSERM UMR_S 1109, ImmunoRhumatologie Moléculaire, Fédération Hospitalo-Universitaire OMICARE, Fédération de Médecine Translationnelle de Strasbourg, Institut d’Immunologie et d’Hématologie, 4 rue Kirschleger, CEDEX, 67085 Strasbourg, France; 8Histocompatibility Laboratory, UMR-S949 Inserm, Etablissement Français du Sang, 67000 Strasbourg, France; anne.parissiadis@efs.sante.fr; 9Nephrology Department, CHU Rouen, 76000 Rouen, France; Isabelle.Etienne@chu-rouen.fr; 10Laboratory of Histocompatibility, EFS Hauts-de-France Normandie, 76000 Rouen, France; Francoise.HAU@efs.sante.fr; 11Service de Néphrologie-Dialyse-Transplantation, CHU de Nice, Université Côte d’Azur, 06000 Nice, France; albano.l@chu-nice.fr; 12Department of Immunology, Centre Hospitalier Universitaire de Nice, University Hospital of Nice, 06000 Nice, France; pourtein.m@chu-nice.fr; 13Department of Renal Transplantation, INSERM U 1082, Sorbonne Université, Pitié-Salpêtrière, 75013 Paris, France; benoit.barrou@aphp.fr; 14Laboratory of Immunology and Histocompatibility, Hôpital Saint-Louis APHP, 75010 Paris, France; jean-luc.taupin@aphp.fr; 15INSERM U976 Institut de Recherche Saint-Louis, Université Paris Diderot, 76013 Paris, France; 16Nephrology Department, CHU de Saint-Etienne, Groupe Immunité Muqueuse et Agents Pathogènes (GIMAP), Jean Monnet University, Université de Lyon, 42100 Saint-Etienne, France; christophe.mariat@univ-st-etienne.fr; 17Laboratoire d’Immunologie, EFS Loire-Auvergne, 25 Boulevard Pasteur, 42100 Saint-Etienne, France; lena.absi@efs-sante.fr; 18Inserm, EHESP, Irset (Institut de recherche en santé, environnement et travail)—UMR_S 1085, Univ Rennes, CHU Rennes, 35000 Rennes, France; cecile.vigneau@chu-rennes.fr; 19HLA-HPA Laboratory, 35016 Rennes, France; virginie.renac@efs.sante.fr; 20Service d’Immunologie et Immunogénétique, Hôpital Pellegrin, CHU de Bordeaux Pellegrin, 33000 Bordeaux, France; line-gwenda.guidicelli@chu-bordeaux.fr (G.G.); jonathan.visentin@chu-bordeaux.fr (J.V.); 21Unité Mixte de Recherche CNRS “Immunoconcept” 5164, 33076 Bordeaux, France; 22Université de Bordeaux, 33000 Bordeaux, France; 23Department of Transplantation, Nephrology and Clinical Immunology, Hospices Civils de Lyon, Edouard Herriot Hospital, 5, place d’Arsonval, 69003 Lyon, France; 24Lyon-Est Medical Faculty, Claude Bernard University (Lyon 1), 8, avenue Rockfeller, 69373 Lyon, France

**Keywords:** adherence, immunosuppression, kidney transplantation, dnDSA, graft survival

## Abstract

Background: Non-adherence with immunosuppressant medication (MNA) fosters development of de novo donor-specific antibodies (*dn*DSA), rejection, and graft failure (GF) in kidney transplant recipients (KTRs). However, there is no simple tool to assess MNA, prospectively. The goal was to monitor MNA and analyze its predictive value for *dn*DSA generation, acute rejection and GF. Methods: We enrolled 301 KTRs in a multicentric French study. MNA was assessed prospectively at 3, 6, 12, and 24 months (M) post-KT, using the Morisky scale. We investigated the association between MNA and occurrence of *dn*DSA at year 2 post transplantation, using logistic regression models and the association between MNA and rejection or graft failure, using Cox multivariable models. Results: The initial percentage of MNA patients was 17.7%, increasing to 34.6% at 24 months. Nineteen patients (8.4%) developed *dn*DSA 2 to 3 years after KT. After adjustment for recipient age, HLA sensitization, HLA mismatches, and maintenance treatment, MNA was associated neither with *dn*DSA occurrence, nor acute rejection. Only cyclosporine use and calcineurin inhibitor (CNI) withdrawal were strongly associated with *dn*DSA and rejection. With a median follow-up of 8.9 years, GF occurred in 87 patients (29.0%). After adjustment for recipient and donor age, CNI trough level, *dn*DSA, and rejection, MNA was not associated with GF. The only parameters associated with GF were *dn*DSA occurrence, and acute rejection. Conclusions: Prospective serial monitoring of MNA using the Morisky scale does not predict *dn*DSA occurrence, rejection or GF in KTRs. In contrast, cyclosporine and CNI withdrawal induce dnDSA and rejection, which lead to GF.

## 1. Introduction

Despite the use of potent oral immunosuppressants, an alloimmune response directed against the donor is the dominant cause of kidney graft failure (GF). This response involves both T cells with borderline and T-cell-mediated rejection (TCMR) and donor-specific antibodies (DSA), which lead to antibody-mediated rejection (ABMR) [1,2]. These inflammatory phases lead to interstitial fibrosis, tubular atrophy (IFTA) [1,3,4,5], transplant glomerulopathy (TG) [6], and finally, GF [7].

The main determinants of *de novo* DSA (*dn*DSA) occurrence in kidney transplant recipients (KTRs) are young recipient age, class II HLA mismatch, high epitope mismatching [8,9,10,11], low calcineurin inhibitor (CNI) trough level [9,12,13,14], CNI withdrawal [15,16], and medication non-adherence (MNA) [1,17]. No effective therapy is available to halt *dn*DSA-induced kidney allograft injury. Therefore, prevention of these injuries is based on HLA matching and early intervention [10,18].

Today, the issue is no longer whether MNA fosters the development of *dn*DSA [8], ABMR [17,19], TCMR [1,20], or GF [21,22,23,24]. Indeed, this has been demonstrated in retrospective studies based on patient reports, suspicion by clinicians, or by the medication possession ratio based on Medicare claims. Yet, there is no approved and simple tool to measure MNA. The CNI trough level can be affected by ‘white-coat adherence’ [25], and CNI variability is not specific for MNA [26,27]. Electronic monitoring is the best method to assess the effects of missed medication [25,28], but its clinical use is difficult and expensive, and prospective studies have yielded opposite outcomes [28,29,30,31,32,33,34]. Prospective self-reported measurement using a questionnaire is a very simple and cost-effective method [35], but the results of prospective studies based on self-reporting are conflicting [20,33,34,36,37].

The usefulness of MNA monitoring by prospective self-reported measurement using a questionnaire in the clinical setting is debated [35,38]. Moreover, there is a lack of granularity in the definition of MNA: the quantity and frequency of missed medication or delay in taking medication sufficient to induce *dn*DSA, a rejection episode, or GF is unknown. The Non-Adherence Consensus Conference in 2008 defined non-adherence as the “deviation from the prescribed medication regimen sufficient to influence adversely the regimen’s intended effect” [38]. The goal of our study was then to determine whether repeated prospective measures of non-adherence using the Morisky questionnaire [39] in KTRs could predict the development of *dn*DSA, rejection, or GF.

## 2. Patients and Methods

### 2.1. Study Design

This study is derived from our previous French multicenter study on MNA after kidney transplantation (KT) [40]. Patients were eligible if they were 18 years or older, French speaking, and underwent KT between 1 August 2006 and 15 March 2009 in one of the participating centers (Bordeaux, Lyon, Nice, Paris La Pitié-Salpétrière, Rennes, Rouen, Saint-Etienne, and Strasbourg hospitals). We excluded patients whose medical files were lost. The flow chart is shown in Figure 1. The KTRs were followed for 10 years post-KT or until GF. All patients gave consent at enrollment in the initial study. The follow-up did not result in any change of care. This study was approved by the ethics committee of the Bordeaux University Hospital (Reference Approval CE-GP-2021-01).

### 2.2. CNI Regimens

Maintenance treatments (tacrolimus, cyclosporine, mycophenolate mofetil [MMF], and steroids), as well as the trough levels of tacrolimus and cyclosporine, were retrospectively collected at 3, 6, 12, and 24 months, post KT. As part of the observational design of the initial study, the CNI targets used were left to the discretion of each center.

The prescribed CNI regimen was defined first according to three categories of maintenance treatment (tacrolimus, cyclosporine or CNI withdrawal). CNI withdrawal was defined by the absence of tacrolimus or cyclosporine on the patient’s prescriptions and was not due to MNA. We also tested different CNI trough levels (among 5, 6, 7, and 8 ng/mL for tacrolimus and 75, 100, 125, and 150 ng/mL for cyclosporine) associated with dnDSA occurrence, rejection, or GF. We also measured the coefficient of variation for the CNI (CNI CV) and the cyclosporin and tacrolimus areas under the curves (AUC) from month 3 to month 24 (including the 4 measurements of CNI trough levels).

### 2.3. Evaluation of Medication Non-Adherence

Adherence was measured using the Morisky scale at 3, 6, 12, and 24 months post-KT, at the time of an outpatient visit, the same day of blood sampling. The questionnaires were collected by the study coordinators. Physicians and nurses were blinded to the responses [40]. This scale has been tested in KTRs and comprises four items related to medication-taking behavior in the prior week [39]. Patients responded “yes” or “no” to each item. The adherence status was estimated at each visit, and non-adherent (NA) patients were defined as those who responded “yes” to at least 1 of the 4 questions according to the Morisky definition. Moreover, for each patient, we determined (1) a level score of NA at each visit, which was the sum of positive answer to each item (score comprised between 0 and 4, 0 being the lowest level of medication adherence and 4 the highest), and (2) a global level score of NA for all visits, which ranged from 0 to 16, and was calculated by summing the scores at each visit. Finally, we estimated the proportion of patients who were always adherent (defined as patients who had a global level score of NA of 16), and patients who were always NA (defined as patient who had a score ≤ 3 at each of the four follow-up visits).

### 2.4. Outcomes

The primary endpoint was the occurrence of *dn*DSA within 2 to 3 years after KT, defined as the occurrence of *de novo* anti-HLA antibodies directed against donor incompatible HLA molecules. We retrospectively collected serum from each patient at the time of KT and at 2 to 3 years after KT to detect *dn*DSA against donors of A, B, Cw, DR, DP and DQ loci. Serum samples were analyzed using Luminex Single-Antigen Flow Beads assays (DSA class I and class II, Lifecodes, Immucor, Norcross, GA). The MFI was measured on a LABscan IS 200, and all specificities with an MFI > 500 and AD-BCR > 5 were considered positive (AD-BCR is MFI adjusted to the quantity of coated antigen per bead). Anti-HLA antibody profiles were analyzed in a blinded fashion by an expert histocompatibility biologist (VD) at the Lyon HLA laboratory.

The secondary endpoints were clinical 5-year biopsy-proved acute rejection and GF. Acute rejection episodes were recorded in medical files if the pathological analysis detected T-cell-mediated rejection (TCMR), borderline (BL), or antibody-mediated rejection (ABMR). To increase the power of our analyses and to take into account patient death, which could be impacted by NA, GF was defined as death, return to dialysis, or retransplantation, whichever came first. These secondary endpoints were retrospectively collected by the same physician (MPR) who was blinded to the adherence status at the time of data collection.

### 2.5. Statistical Analysis

Patients’ characteristics are shown as median interquartile ranges (IQRs) for quantitative variables and as percentages for qualitative variables. At each time of MNA and CNI trough level measurement, we compared the CNI trough level between adherent and non-adherent patients levels using a Student’s *t*-test.

We estimated the effect of MNA at 3, 6, 12, and 24 months and the effect of CNI regimens at the same time on the induction of *dn*DSA, using logistic regression models. Patients without available serum samples were excluded from theses analyses.

We estimated the effect of MNA and CNI regimens on acute rejection by a cause-specific proportional hazard model censored at the time of GF, which was considered a competing event. GF that occurred before the time of measuring CNI or MNA were excluded.

We estimated the probability of survival with a functioning graft at 10 years post-KT according to adherence status and CNI regimen, using the Kaplan–Meier estimator and the log-rank test to compare the hazard of GF. We estimated the effect of MNA and CNI regimens on GF using Cox proportional hazard regressions.

To assess the effect of MNA on each outcome, we evaluated MNA according to the adherence status at each time of measure (binary variables) and the global Morisky score at the four visits (continuous variable).

For the survival analyses, separate analyses were performed for each variable of NA presented above. For univariate analyses, the time axis elapsed since the time of measure of MNA and CNI regimen (3, 6, 12 and 24 months) and since the second-year post transplantation for the global Morisky score, including the four visits. For multivariate analyses, the time axis elapsed since the third-year post transplantation because we adjusted for occurrence of *dn*DSA between 2 and 3 years post-transplantation. Finally, we performed a sensitivity Cox analysis with the NA status and CNI regimen (presence or absence of CNI in the treatment) as time-dependent variables to study their impact on graft failure. The time axis for this analysis was the time elapsed since transplantation.

To take into account correlations between failures of patients in the same center, we used a frailty model for each multivariable Cox model. The log-linearity of the effects of all quantitative variables was checked using splines [41], and proportional hazard assumptions were checked using Schoenfeld residuals. The statistical analysis was performed using SAS software ver. 9.4 (SAS Institute Inc., Cary, NC, USA).

## 3. Results

### 3.1. Patients’ Characteristics and Immunosuppression

In total, 301 of the 312 patients from the initial cohort were included in this study. The patients’ characteristics are listed in Table 1. At inclusion, 75.1% of patients received tacrolimus, 24.9% cyclosporine, and 97.3% MMF. These percentages were stable over time. The percentage of patients that underwent CNI withdrawal (replaced by mTOR inhibitors) increased from 2.3% at 3 months post-KT to 5.9% at 24 months. Finally, the percentage of patients receiving steroids was 91.3% at inclusion and decreased to 49.7% at 24 months (Table 2).

### 3.2. Medication Non-Adherence

The percentage of MNA patients (defined as at least one response of “yes” to one of the four questions at any given visit) increased after KT to 17.6% at 3 months, 23.6% at 6 months, 31.2% at 12 months, and 34.6% at 24 months (Table 2). At each visit, less than 6% of patients had a score of ≤2, showing than most of the MNA patients were NA regarding only one item. The majority of positive responses concerned forgetting to take medication and/or when to take immunosuppressants.

The median global MNA score for all visits was 15 [14,15,16] on a scale of 0–16. At 2 years post-KT, 48.7% of patients were always adherent and only 12 (4%) were always NA (Table 2). We did not find any association between MNA measured using the Morisky scale at each visit and the CNI trough levels (Appendix A).

### 3.3. De Novo Donor-Specific Antibodies

The results of the anti-HLA antibody screening were available for 226 patients at 30 ± 6 months post-KT. At 30 months, 19 patients (8.4%) developed *dn*DSA. In total, 25 *dn*DSA (15 anti-DQ, 6 anti-DR, 1 anti-A and 3 anti-B) were detected with a median MFI of 5650 (IQR 25–75 3425–14,725).

Factors associated with *dn*DSA were younger recipient age, cyclosporine use, and CNI withdrawal (Table 1). Irrespective of when the CNI level was measured, we did not find any association between CNI through level or CV and occurrence of *dn*DSA.

After adjustment for recipient age, baseline sensitization, HLA mismatch, and maintenance treatment, *dn*DSA occurrence did not differ between adherent and NA patients at 3 months (OR 1.16, 95% CI 0.31–4.45, *p*-value = 0.73), 6 months (OR 1.40, 95% CI 0.43–4.56, *p*-value = 0.44), 12 months (OR 0.48, 95% CI 0.14–1.66, *p*-value = 0.31), and 24 months (OR 2.01, 95% CI 0.59–6.83, *p*-value = 0.19) (Table 3). A decrease of one point on the global Morisky score of the four visits was not associated with *dn*DSA occurrence (OR 1.14, 95% CI 0.77–1.69, *p*-value = 0.80). In these multivariate models, only cyclosporine use at 3, 12, and 24 months and CNI withdrawal at 6, 12, and 24 months were associated with *dn*DSA occurrence.

### 3.4. Rejection

Seventy-three patients (24.2%) suffered at least one episode of biopsy-proven acute rejection during the first 5 years post-KT. The first episode was a BL rejection in 15 patients (5.0%, median time of occurrence: 5.7 months post-KT), an acute TCMR in 43 patients (14.2%, 4.7 months), and an active ABMR in 15 patients (5.0%, 18.7 months). At 2 to 3 years post-KT, *dn*DSA was found in 4.2% of patients without rejection, 10.0% with BL rejection, 16.2% with TCMR, and 38.5% with ABMR (*p*-value of a Fisher test = 0.0003).

We did not analyze ABMR separately due to the small number of events. Among the patients’ characteristics, only cyclosporine use and CNI withdrawal were associated with at least one rejection episode (including BL rejection, TCMR, or ABMR) (Appendix A).

After adjustment for CNI use only, we did not find an association between 5-year acute rejection and MNA at 3 months (HR 0.93, 95% CI 0.40–2.15, *p*-value = 0.94), 6 months (HR 1.19, 95% CI 0.52–2.75, *p*-value = 0.77), 12 months (HR 0.52, 95% CI 0.17–1.59, *p*-value = 0.27), or 24 months (HR 0.57, 95% CI 0.11–2.91, *p*-value = 0.60) (Appendix A). In these multivariate models, only cyclosporine use at 3 and 12 months, and CNI withdrawal at 3, 6 and 12 months were associated with rejection.

### 3.5. Graft Survival

Finally, during a median follow-up of 8.9 years (IQR 25–75 8.0–9.8), GF occurred in 87 patients (29.0%) (49 returns to dialysis and 38 deaths). GF was associated with donor age and cyclosporine use at 3 and 6 months (Appendix A). After testing different CNI trough levels, we found that patients with tacrolimus trough levels below 5 ng/mL and cyclosporine trough levels below 100 ng/mL at 24 months post-transplant (defined as patients with low CNI through levels, *n* = 44) had a probability to survive with a functioning graft for ≥5 years post-KT of 76.1% vs. 95.1% for patients with tacrolimus trough levels above 5 ng/mL and cyclosporine trough levels above 100 ng/mL at 24 months post-transplant (defined as patients with standard CNI regimen) (*n* = 188) (Figure 2 and Appendix A). Likewise, patients who had a graft failure had a median tacrolimus AUC lower than patients without graft failure.The probability to survive with a functioning graft for ≥5 years post-KT was also lower (72.7%) among those with CNI withdrawal (*n* = 20). In addition, GF was associated with *dn*DSA and rejection (Appendix A).

After adjustment for recipient and donor age, CNI regimen, occurrence of *dn*DSA between 2 and 3 years, and acute rejection (using a time-dependent variable), we did not find an association between GF and MNA at 3 months (HR 1.07, 95% CI 0.52–2.24, *p*-value = 0.84), 6 months (HR 1.17 95% CI 0.58–2.34, *p*-value = 0.73), 12 months (HR 0.92, 95% CI 0.48–1.78, *p*-value = 0.46), and 24 months (HR 0.96, 95% CI 0.50–1.85, *p*-value = 0.92) (Table 4). A decrease of one point in the global Morisky score was not associated with an increase in the hazard of GF (HR 1.01, 95% CI 0.82–1.24, *p*-value = 0.80). Furthermore, patients who were NA for 24 months post-KT did not have a significant increase in the hazard of GF compared to patients who were adherent for 24 months (HR 1.72, 95% CI 0.48–6.16, *p*-value = 0.67). The results were unchanged when NA status was analyzed as a time-dependent variable.

In the second model, the CNI regimen (tacrolimus vs. cyclosporine) was replaced by CNI trough level (standard vs. low CNI trough levels), but there was still no association between GF and MNA (Appendix A). After adjustment for recipient and donor age, occurrence of *dn*DSA and acute rejection, the CNI trough level at 24 months was no longer associated with GF (Appendix A). In these two models, GF was mainly driven by *dn*DSA and acute rejection, which were the only factors associated with GF.

## 4. Discussion

This prospective and multicenter study with a 10-year follow-up was an attempt to validate the Morisky questionnaire with clinical endpoints in the field of transplantation. However, it was not able to show any relationships between repeated measures of adherence with this scale and *dn*DSA occurrence, further rejection, or GF. In this cohort, GF was mainly driven by *dn*DSA occurrence and/or rejection. These two features reflect the induction of a donor-specific alloimmune response, which was observed mainly in patients on cyclosporine or who underwent CNI withdrawal.

In previous studies using self-reports, Vlaminck et al. found a higher rate of late rejection in MNA patients [20]. Despite repeated measures of MNA with the Morisky questionnaire, we did not confirm this finding, which had been previously challenged in the Swiss SMART study [33,34]. Overall, we usually postulate that questionnaires result in overestimation of adherence because results are often distorted by patients’ unwillingness to report inappropriate behavior [24]. In our study, physicians and nurses were blinded to the questionnaire responses, which could have limited this phenomenon. Moreover, MNA is often associated with reluctance to attend consultation appointments [42], thus rendering diagnosis of MNA by questionnaire impossible. Notably, MNA was defined as forgetting to take medication or a mistake in the time of taking immunosuppressive drugs in the majority of the patients, implying that isolated omission or mistakes in the timing of a dose do not negatively impact the outcome of KT. Finally, the Morisky questionnaire is not the most appropriate tool because it does not take into account when the immunosuppressants are taken [43]. For all these reasons, we propose that repeated and prospective use of the Morisky questionnaire should not be recommended for tracking MNA in KTRs. On the contrary, tools collecting patients’ medications beliefs and illness perceptions could help to better understand the reasons of MNA.

Other tools should be tested in prospective studies, but all have drawbacks. The within-patient variability in a tacrolimus trough level of >30% may be a marker of *dn*DSA occurrence, late rejection, transplant glomerulopathy, progression to severe IFTA, and GF [26,27,44,45]. However, it is not specific to MNA [25,26]. Electronic monitoring is expensive and is not available for all patients. Calculating the medication possession ratio is also feasible [20,21], but such data are not available to physicians in real-time. Finally, combining self-reporting, CNI trough levels and clinicians’ reports has the best sensitivity and specificity for evaluating MNA [38,46]; however, in our study, it was not possible to determine whether low CNI trough levels were due to MNA or physician prescriptions. Therefore, despite its dramatic consequences, MNA is misdiagnosed due to the lack of an easy and reliable tool.

Patients receiving cyclosporine during the first 2 years post-KT were more prone to developing *dn*DSA [9,47], and rejection [48]. In our cohort, and as reported previously, CNI withdrawal was also associated with an increased frequency of *dn*DSA and rejection [15,16]. Unfortunately, we did not assess the reasons for replacement of CNI by mTOR inhibitors, but at the time of this study, this strategy was used during the first 2 years post-KT to prevent the renal toxicity of CNI. A significant number of patients had reduced exposure to CNI at 24 months post-KT. We were unable to determine whether this was due to a medical prescription or MNA. It is important to note that at the time of this study (2006–2009), reduced exposure to CNI could have been caused by the physician, as CNI toxicity was considered the main cause of GF [49]. In recent studies, CNI trough levels of <3–7 ng/mL for tacrolimus and <75–100 ng/mL for cyclosporine have been associated with *dn*DSA occurrence [8,12,13,14]. Despite testing several CNI trough levels, we were unable to determine a level of tacrolimus or cyclosporine associated with *dn*DSA occurrence or rejection in this cohort. The priming of an alloimmune response sufficient to trigger *dn*DSA involves factors other than immunosuppression, such as a mismatched epitope load provided by the donor [9,11], or the presence of danger signals induced, for instance, by ischemia and reperfusion.

The main limitation of this study was inherent in its prospective design. Patients who participate in such a study are more likely to be adherent, and it can be assumed that the least-adherent patients did not agree to participate. As a consequence, the percentage of GF in this study was lower than in the global French transplant population, and the negative results could be due to a lack of power. However, if the least adherent patients declined to complete the questionnaire in the context of a clinical study, they are likely to also be unwilling to do so in the clinical setting, making this tool inappropriate.

In conclusion, MNA determined by the Morisky scale was not associated with subsequent *dn*DSA occurrence, rejection, and GF in this cohort, but CNI withdrawal and cyclosporine were, as previously reported. New tools are therefore needed for measuring MNA prospectively.

The English in this document has been checked by at least two professional editors, both native speakers of English.

## Figures and Tables

**Figure 1 jcm-10-02032-f001:**
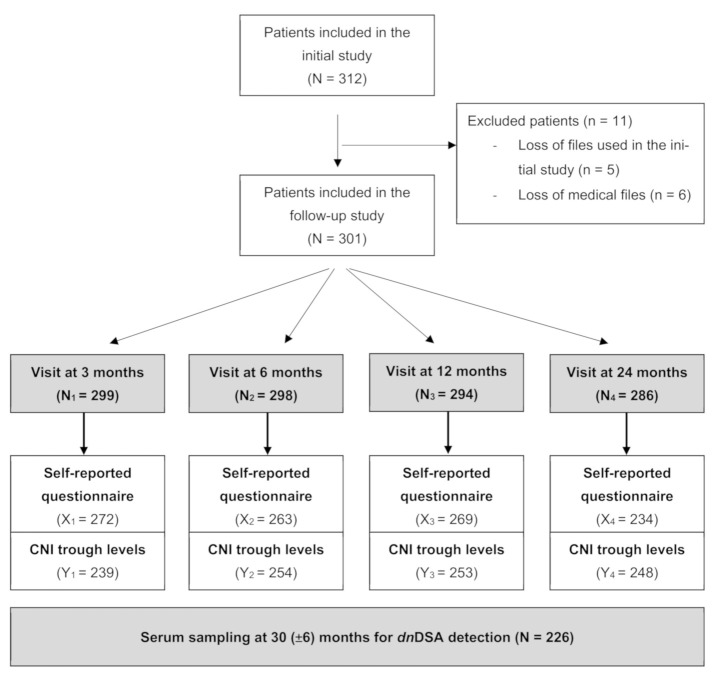
Study design flow chart. Patients completed adherence self-questionnaires at months 3, 6, 12, 24. CNI trough levels were collected retrospectively at months 3, 6, 12 and 24 (±30 days). Serum samples were collected retrospectively between 2 and 3 years post transplantation for detecting dnDSA. At each visit, some data were available for N_x_ patients. Among these N_x_ patients, X_x_ patients responded to the questionnaire and CNI through levels were available for Y_x_ patients.

**Figure 2 jcm-10-02032-f002:**
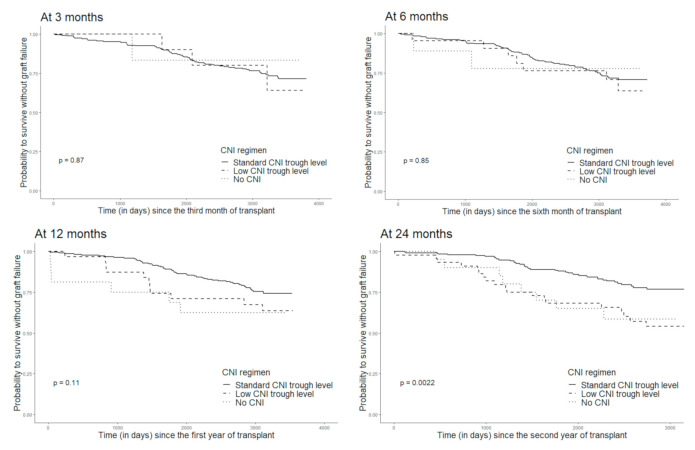
Probability of surviving without graft failure according to the CNI trough level at 3, 6, 12 and 24 months post transplantation (Kaplan–Meier estimator).

**Table 1 jcm-10-02032-t001:** Patients’ characteristics at time of transplantation.

	Whole Cohort		By Occurrence of *dn*DSA between 2 and 3 Years Post Transplant			
	N		N	No dnDSA *n* = 207	dnDSA *n* = 19	*p* Value
**Age** (years), median (IQR)	301	51.0 (40.0–60.0)	226	48.8	42.3	0.04
**Male**, *n* (%)	301	208 (69.1)	226	145 (70.0)	15 (79.0)	0.60
**Primary kidney disease**	295		221			0.42
Vascular, *n* (%)		10 (3.4)		6 (3.0)	0 (0.0)	
Diabetics, *n* (%)		24 (8.1)		16 (7.9)	2 (10.5)	
Glomerulonephritis, *n* (%)		82 (27.8)		58 (28.7)	9 (47.4)	
Tubulo-interstitial, *n* (%)		31 (10.5)		24 (11.9)	3 (15.8)	
Hereditary (including polycystic kidney disease), *n* (%)		64 (21.7)		52 (25.7)	2 (10.5)	
Other or undetermined, *n* (%)		84 (28.5)		46 (22.8)	3 (15.8)	
**Dialysis modality before transplant**	292		203			0.11
Preemptive transplantation, *n* (%)		27 (9.3)		1 (0.5)	0 (0.0)	
Hemodialysis, *n* (%)		221 (75.7)		147 (79.5)	18 (100.0)	
Peritoneal dialysis, *n* (%)		44 (15.1)		37 (0.2)	0 (0.0)	
**Number of previous graft(s)**	271		224			0.36
0, *n* (%)		224 (82.7)		172 (83.9)	16 (84.2)	
1, *n* (%)		41 (15.1)		30 (14.6)	2 (10.5)	
≥2, *n* (%)		6 (2.3)		2 (1.0)	1 (5.3)	
**Sensitized patients**, *n* (%)	226	61 (27)	226			0.11
Yes				59 (28.5)	2 (10.5)	
No				148 (71.5)	17 (89.5)	
**Donor age** (years), mean (SD)	278	49.2 (15.9)	217	47.1 (15.5)	45.8 (18.4)	0.75
**Living donor**, *n* (%)	298	9 (3.0)	225	6 (2.9)	0 (0.0)	1.00
**Expanded criteria donor**, *n* (%)	274	116 (42.3)	213	67 (34.5)	5 (26.3)	0.61
**Number of HLA mismatches (A–B–DR)**	295		224			0.12
0, *n* (%)		3 (1.0)		1 (0.5)	0 (0.0)	
1, *n* (%)		10 (3.4)		7 (3.4)	0 (0.0)	
2, *n* (%)		26 (8.8)		14 (6.8)	1 (5.3)	
3, *n* (%)		48 (16.3)		29 (14.1)	4 (21.1)	
4, *n* (%)		91 (30.8)		64 (31.2)	3 (15.8)	
5, *n* (%)		85 (28.8)		61 (29.8)	11 (57.9)	
6, *n* (%)		32 (10.8)		29 (14.1)	0 (0.0)	
**Total ischemia time** (hours), median (IQR)	269	15.8 (13.0–19.0)	212	16.5 (6.0)	16.2 (6.8)	0.84
**Induction treatment**	276		206			0.41
Basiliximab, *n* (%)		168 (60.9)		125 (66.1)	9 (52.9)	
rATG, *n* (%)		108 (39.1)		64 (33.9)	8 (47.1)	
**Delayed graft function**, *n* (%)	292	61 (20.9)	220	40 (19.8)	5 (27.8)	0.38
**Maintenance treatment at 3 months post-transplant**	296		219			0.002
Cyclosporine		64 (21.6)		46 (23.0)	11 (57.9)	
Tacrolimus		227 (76.7)		151 (75.5)	7 (36.8)	
Without CNI		5 (1.7)		3 (1.5)	1 (5.3)	
**Maintenance treatment at 6 months post-transplant**	294		219			0.004
Cyclosporine		61 (20.7)		46 (23.0)	9 (47.4)	
Tacrolimus		226 (76.9)		150 (75.0))	8 (42.1)	
Without CNI		7 (2.4)		4 (2.0)	2 (10.5)	
**Maintenance treatment at 12 months post-transplant**	291		219			0.02
Cyclosporine		55 (18.9)		42 (21.0)	8 (42.1)	
Tacrolimus		224 (77.0)		151 (75.5)	9 (47.4)	
Without CNI		12 (4.1)		7 (3.5)	2 (10.5)	
**Maintenance treatment at 24 months post-transplant**	282		218			0.01
Cyclosporine		53 (18.8)		41 (20.6)	7 (36.8)	
Tacrolimus		214 (75.9)		150 (75.4)	9 (47.4)	
Without CNI		15 (5.3)		8 (4.0)	3 (15.8)	
**Corticosteroids at 3 months**	301		226			0.23
Continued		255 (84.7)		190 (91.8)	16 (84.2)	
Discontinued		46 (15.3)		17 (8.2)	3 (15.8)	

N: subjects with available data; IQR: Interquartile range; SD: standard deviation; HLA: Human Leucocyte Antigen; CMV: Cytomegalovirus. The bold elements correspond to the headers and differentiate the name of the variable from the categories of this variable.

**Table 2 jcm-10-02032-t002:** Immunosuppression maintenance treatment and medication non-adherence assessed by the Morisky scale over time.

	**Follow-Up Visits**			
	**Immunosuppression Maintenance Treatment**			
	**M3 (N** **= 299)**	**M6 (N** **= 298)**	**M12 (N** **= 294)**	**M24 (N** **= 286)**
Tacrolimus, *n* (%)	227 (75.9)	228 (76.5)	225 (76.5)	216 (75.5)
Cyclosporine, *n* (%)	65 (21.7)	61 (20.5)	55 (18.7)	53 (18.5)
MMF, *n* (%)	283 (94.6)	266 (89.2)	263 (89.4)	251 (87.8)
Steroids, *n* (%)	255 (85.3)	203 (68.1)	166 (56.5)	142 (49.7)
CNI withdrawal, *n* (%)	7 (2.3)	9 (3.0)	14 (4.8)	17 (5.9)
Trough levels of tacrolimus, med (IQR)	8.8 (7.3–10.8)	8.2 (6.7–9.7)	7.6 (6.4–9.3)	7.5 (6.2–8.9)
Trough levels of cyclosporine, med (IQR)	156 (130–190)	149 (112–171)	128.5 (96–157)	93 (65–125)
	**Treatment Medication Non-Adherence Assessed by the Morisky Scale**			
	**M3 (N = 272)**	**M6 (N** **= 263)**	**M12 (N** **= 269)**	**M24 (N** **= 234)**
**Positive response to questions of the Morisky scale**, *n* (%)				
Do you ever forget to take your medicine?	19 (7.0)	29 (10.9)	40 (14.9)	46 (19.5)
Are you careless at times about taking your medicine?	33 (12.1)	40 (15.2)	54 (20.1)	49 (20.8)
When you feel better, do you sometimes stop taking your medicine?	2 (0.7)	0 (0.0)	2 (0.7)	1 (0.4)
Sometimes if you feel worse when you take your medicine, do you stop taking it?	6 (2.2)	2 (0.7)	3 (1.1)	2 (0.8)
**Non-adherent patients** ^†^, *n* (%)	48 (17.6)	62 (23.6)	84 (31.2)	81 (34.6)
**Always non-adherent patients** ^‡^, *n* (%)	12 (4.0)			
**Score of the Morisky scale**, *n* (%)				
4	224 (82.4)	201 (76.4)	185 (68.8)	153 (65.4)
3	41 (15.1)	54 (20.5)	72 (26.8)	66 (28.2)
≤2	7 (2.6)	8 (3.1)	12 (4.4)	15 (6.4)
**Global score ***, *n* (%)				
16	96 (48.7)			
15	42 (21.3)			
13–14	42 (21.3)			
≤12	17 (8.6)			

M = month; N = subjects with available data; MMF = mycophenolate mofetil; med = median; IQR = interquartile range. ^†^ subjects with at least one positive response at Morisky scale; ^‡^ subjects without missing data with a positive response at each visit; * sum of positive response(s) of every Morisky scale at each time point for patients without missing data.

**Table 3 jcm-10-02032-t003:** Effect of non-adherence measured by the Morisky scale and immunosuppression regimen on the risk of *dn*DSA apparition between 2 and 3 years post-transplant among patients with an available serum sample. Results of logistic regression model. Multivariable analysis.

	3 Months (N ^‡^ = 200)			6 Months (N ^‡^ = 193)			12 Months (N ^‡^ = 198)			24 Months (N ^‡^ = 174)		
	OR *	CI 95%	*p*-Value	OR *	CI 95%	*p*-Value	OR *	CI 95%	*p*-Value	OR *	CI 95%	*p*-Value
**Non adherent vs. adherent** ^†^	1.16	0.31–4.45	0.73	1.40	0.43–4.56	0.44	0.48	0.14–1.66	0.31	2.01	0.59–6.83	0.19
**Maintenance treatment**			0.005			0.04			0.009			0.005
Tacrolimus	1			1			1			1		
*Cyclosporine*	5.2	1.65–16.4		2.49	0.76–8.16		4.67	1.48–14.76		5.17	1.34–20.01	
Without anticalcineurin	15.3	0.84–279.6		15.5	1.46–165.2		7.39	1.08–50.47		11.8	1.34–69.6	
**Recipient age** (per year)	0.97	0.93–1.01	0.06	0.96	0.92–0.99	0.02	0.96	0.92–0.99	0.03	0.97	0.93–1.02	0.15
**HLA mismatch** (per number of mismatch)	1.04	0.66–1.64	0.75	1.06	0.67–1.67	0.62	1.02	0.66–1.58	0.91	0.94	0.59–1.50	0.96
**Sensitized patients** vs. non-sensitized patients	0.21	0.03–1.72	0.91	0.21	0.03–1.74	0.95	0.18	0.02–1.42	0.78	0.21	0.03–1.74	0.81

OR: Odds ratio; CI: Confidence Interval; ^‡^ Subjects with complete data and with functioning graft at time of measure of NA; ^†^ Subjects with at least one positive response to Morisky scale at time of measure. * adjusted for adherence status, maintenance treatment, recipient age, HLA mismatch and sensitization at inclusion.

**Table 4 jcm-10-02032-t004:** Effect of non-adherence measured by the Morisky scale and immunosuppression regimen (tacrolimus/cyclosporine/any anticalcineurin) on the hazard of graft failure since the third year post transplantation. Results of Cox proportional hazard model. Multivariable analysis.

	3 Months (N ^‡^ = 190)			6 Months (N ^‡^ = 183)			12 Months (N ^‡^ = 191)			24 Months (N ^‡^ = 169)		
	HR *	CI 95%	*p*-Value	HR *	CI 95%	*p*-Value	HR *	CI 95%	*p*-Value	HR *	CI 95%	*p*-Value
**Non adherent vs. adherent** ^†^	0.89	0.40–2.00	0.84	1.08	0.53–2.23	0.73	0.97	0.50–1.88	0.46	0.93	0.49–1.79	0.92
**Recipient age** (per year)	1.01	0.98–1.05	0.78	1.01	0.97–1.04	0.63	1.01	0.97–1.04	0.75	1.00	0.97–1.04	0.72
**Donor age**			0.68			0.91			0.71			0.24
<38 years	1			1			1			1		
39–50 years	1.34	0.51–3.53		1.21	0.44–3.35		1.10	0.42–2.93		1.92	0.63–5.85	
51–59 years	1.80	0.67–4.83		1.54	0.54–4.40		1.49	0.54–4.10		2.73	0.84–8.85	
≥60 years	2.01	0.62–6.60		1.04	0.28–3.87		1.27	0.37–4.38		3.67	0.97–13.9	
**Maintenance treatment**			0.37			0.27			0.97			0.15
Tacrolimus	1			1			1			1		
Cyclosporine	0.90	0.46–1.78		1.30	0.69–2.45		0.86	0.45–1.65		0.76	0.37–1.56	
Any anticalcineurin	0.46	0.06–3.56		0.26	0.03–2.02		0.71	0.21–2.38		1.20	0.45–3.23	
**Occurrence of acute rejection** (time-dependent variable)	2.64	1.34–5.16	0.0005	2.65	1.43–4.90	0.002	1.92	1.06–3.49	0.03	1.90	0.98–3.67	0.055
**Occurrence of *dn*DSA** between 2 and 3 years post-transplant	2.42	1.02–5.75	0.03	3.51	1.49–8.25	0.002	2.30	1.03–5.12	0.04	1.84	0.74–5.23	0.19

HR: Hazard ratio; CI: Confidence Interval; ^‡^ Subjects with complete data and with a functioning graft at 3 years post-transplant; ^†^ Subjects with at least one positive response to Morisky scale at time of measure. * adjusted for recipient and donor age, maintenance treatment at time of measure of NA, occurrence of acute rejection (time dependent variable) and occurrence of dnDSA between 2 and 3 years post transplant.

## Data Availability

The data underlying this article will be shared upon reasonable request to the corresponding author.

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
