# Peer review of "Prospective Measures of Adherence by Questionnaire, Low Immunosuppression and Graft Outcome in Kidney Transplantation"

_jcm, 2021, doi:10.3390/jcm10092032_

Round 1

Reviewer 1 Report

Immunosuppressive therapy non-adherence is a well-established cause of de novo DSA development and graft loss. You analyze very well in the discussion section the reasons why assessing non-adherence via a questionnaire is not efficient.

It is also well-known, that CNI withdrawal and cyclosporine are also associated with graft rejection. It is not very clear in the study if CNI withdrawal was due to non-adherence or not and if evaluating this factor was in the objectives of the study. 

Author Response

Thank you for your comments.

In our study, we recorded the treatments prescribed by transplant physicians from patient’s medical files. CNI withdrawal was defined by the absence of tacrolimus or cyclosporine on the patient’s prescriptions. Therefore, CNI withdrawal was not due to non-adherence. To clarify this point, we added this sentence, page 4, lines 24-26: ‘CNI withdrawal was defined by the absence of tacrolimus or cyclosporine on the patient’s prescriptions and was not due to MNA.’

Reviewer 2 Report

Nonadherence is very important. We know from the paper of Selares and Halloran with coauthors that nonadherence is the main cause of antibody mediated rejection and graft loss.

Clinicians need appropriate tool to measure nonadherence.

Mathilde Prezelin-Reydit and other authors from different French centres in “Prospective measures of adherence by questionnaire, low immunosuppression and graft outcome in kidney transplantation” tried to verified Morisky questionnaire to predict dnDSA occurrence, rejection or graft failure in patients after kidney transplantation.

They concluded that nonadherence with immunosuppressant medication using questionnaire does not predict dn DSA occurrence, rejection or graft failure.

Authors analyze only Morisky questionnaire with few questions so maybe it should be pointed in conclusions of abstract.

In conclusions after discussion the first is written: CNI withdrawal and cyclosporine were associated with subsequent dnDSA occurrence, rejection and GF in this cohort.

It is well known knowledge.

The second statement is : MNA determined by the Morisky scale was not.

Maybe the paper should focus on Morisky scale only or additionally on the other additional factors which are known and have influence on graft failure.

I was not able to find Figure 2

Author Response

1. Authors analyze only Morisky questionnaire with few questions so maybe it should be pointed in conclusions of abstract.

Thank you for your proposal. We pointed in conclusions of the abstract that NA was evaluated with Morisky scale.

2. In conclusions after discussion the first is written: CNI withdrawal and cyclosporine were associated with subsequent dnDSA occurrence, rejection and GF in this cohort.

It is well known knowledge.

The second statement is: MNA determined by the Morisky scale was not.

Maybe the paper should focus on Morisky scale only or additionally on the other additional factors which are known and have influence on graft failure.

Thank you for your comments. Our paper focused mainly on MNA and its impact on the development of dnDSA, rejection, or graft failure. It was the objective of the study, and our statistical analyses were conducted to answer this question. We fully agree that our main results concern the MNA. We then modified the conclusion as follow to highlight our findings on the Morisky scale: ‘In conclusion, MNA determined by the Morisky scale was not associated with subsequent dnDSA occurrence, rejection, and GF in this cohort. On the opposite, CNI withdrawal and cyclosporine were, as previously reported. New tools are therefore needed for measuring MNA prospectively’

3. I was not able to find Figure 2.

We are sorry for this omission. The Figure 2 was added to the manuscript.

Reviewer 3 Report

The Authors aimed to define a possible relationship between adherence to immunosuppressant medication and the development of dnDSA; in this prospective study the serial monitoring of MNA using questionnaire did not predict dnDSA occurrence, rejection and graft failure in KT recipients. 

The Morisky questionnaire has not been validated in KT recipients, although it can be potentially useful in clinical practice for its short form in evaluating compliance to therapy in KT patients, especially in younger ages. 

The authors performed also an analysis of other risk factor for rejection and graft failure in this cohort.

Abstract

I would suggest to specify the questionnaire that is going to be used/validated to make it clearer.

Introduction

The background has a good flow; the aim of the study is clear.

Methods

page 5 line 3, I would suggest to specify the modality of administration of the test, whether it was administered in hospital/home, at time of scheduled visit, in the same day of blood sampling, in presence or not of the medical doctor; whether the the doctor was blinded or not to the patient's questionnaire results... If this is reported in another previous study of the group, please specify it. These are all data that can change the interpretation of results (e.g. overestimation of patient adherence if the patient knew the doctor can know the results...). These aspects have been also rightly cited by authors (page 10 line 26).

page 4 line 10, please check the reference

Results

this section could be improved with other comments about the interpretation of the questionnaire according to previous suggestion; eventually the modality of administration that can potentially justify an overestimation of the adherence (e.g. blinding the doctors and send the questionnaires to another centre could increase the participation of unwilling patients...).

Author Response

1. Abstract

I would suggest to specify the questionnaire that is going to be used/validated to make it clearer.

Thank you for your suggestion. We specified the Morisky scale in the abstract.

2. Introduction

The background has a good flow; the aim of the study is clear.

Thank you

3. Methods

page 5 line 3, I would suggest to specify the modality of administration of the test, whether it was administered in hospital/home, at time of scheduled visit, in the same day of blood sampling, in presence or not of the medical doctor; whether the the doctor was blinded or not to the patient's questionnaire results... If this is reported in another previous study of the group, please specify it. These are all data that can change the interpretation of results (e.g. overestimation of patient adherence if the patient knew the doctor can know the results...). These aspects have been also rightly cited by authors (page 10 line 26).

Thank you for your comments. We clarified these points in the Methods section, page 5, lines 5-7 as follow: ‘Adherence was measured using the Morisky scale at 3, 6, 12, and 24 months post-KT, at the time of an outpatient visit, the same day of blood sampling. The questionnaires were collected by the study coordinators. Physicians and nurses were blinded to the responses (40)’.

4. page 4 line 10, please check the reference

Thank you for your careful reading. We modified this reference in the revised manuscript.

5. Results

this section could be improved with other comments about the interpretation of the questionnaire according to previous suggestion; eventually the modality of administration that can potentially justify an overestimation of the adherence (e.g. blinding the doctors and send the questionnaires to another centre could increase the participation of unwilling patients...).

Thank for this relevant comment. We added page 11, lines 6-7, some words explaining that the modality of collection of the Morisky scale in our study (Physicians and nurses blinded to the responses) could have limited the overestimation of adherence.

Round 2

Reviewer 2 Report

I accept the paper in present form